# Challenges in Defining a Reference Set of Differentially Expressed lncRNAs in Ulcerative Colitis by Meta-Analysis

Christopher G. Fenton [1,2], Mithlesh Kumar Ray [1] and Ruth H. Paulssen [1,2,*]

1 Clinical Bioinformatics Research Group, Department of Clinical Medicine, UiT-The Arctic University of Norway, N-9037 Tromsø, Norway; christopher.fenton@uit.no (C.G.F.); mithlesh.ray@uit.no (M.K.R.)
2 Genomic Support Centre Tromsø (GSCT), Department of Clinical Medicine, UiT-The Arctic University of Norway, N-9037 Tromsø, Norway
* Correspondence: ruth.h.paulssen@uit.no; Tel: +47-77645480

**Abstract:** The study aimed to identify common differentially expressed lncRNAs from manually curated ulcerative colitis (UC) gene expression omnibus (GEO) datasets. Nine UC transcriptomic datasets of clearly annotated human colonic biopsies were included in the study. The datasets were manually curated to select active UC samples and controls. R packages geneknitR, gprofiler, clusterProfiler were used for gene symbol annotation. The R EdgeR package was used to analyze differential expression. This resulted in a total of nineteen lncRNAs that were differentially expressed in at least three datasets of the nine GEO datasets. Several of the differentially expressed lncRNAs found in UC were associated with promoting colorectal cancer (CRC) through regulating gene expression, epithelial to mesenchymal transition (EMT), cell cycle progression, and by promoting tumor proliferation, invasion, and migration. The expression of several lncRNAs varied between disease states and tissue locations within the same disease state. The identified differentially expressed lncRNAs may function as general markers for active UC independent of biopsy location, age, gender, or treatment, thereby representing a comparative resource for future comparisons using available GEO UC datasets.

**Keywords:** meta-analysis; LncRNAs; ulcerative colitis

## 1. Introduction

The term lncRNA is defined as a non-coding transcript greater than 200 nucleotides in size that does not have the potential to code for a protein. LncRNAs have been shown to directly interact with chromatin-modifying enzymes and nucleosome-remodeling factors to control chromatin structure and accessibility [1]. LncRNAs can regulate transcription of neighboring and distant genes through interacting with DNA, RNA, and proteins [2]. Compared to protein-coding genes, lncRNAs exhibit greater tissue specificity [3]. In recent years, the regulation of long non-coding RNAs (lncRNAs) has been associated with cancer and other diseases [4], yet working with lncRNAs remains challenging. LncRNAs have a low abundance compared with protein coding RNAs, which makes it difficult to separate lncRNA expression from background [5] transcriptional noise [6]. The function of the majority of lncRNAs is unknown [7], and the expression of lncRNA expression may be directly influenced by tissue type [8]. The number of annotated lncRNAs differs vastly between lncRNA databases such as FANTOM, NONCODE, LNCipedia, and others, and the overlap between these lncRNA databases is low [9]. LncRNAs have been recognized as key players in many diseases, including ulcerative colitis (UC) [5,10].

UC is a chronic relapsing–remitting inflammatory disease of the gastrointestinal tract that is associated with genetics, the host immune system, and environmental factors [11]. Chronic inflammation in UC has been shown to increase the risk for the development of colorectal cancer (CRC) [12]. Unfortunately, the pathophysiology of UC is still unclear. The status of inflammation and grade of severity are usually determined by clinical, histologic,

endoscopic, and laboratory parameters [13–17]. Currently, the gold standard for the diagnosis of UC is endoscopy [14,16]. Moreover, many UC patients experience relapses eventually [18,19]. Therefore, it is important to improve UC prognosis and diagnosis through a more thorough molecular characterization which will pave the way for more UC-specific therapeutic options.

The precise molecular mechanisms underlying disease UC pathogenesis remain elusive despite significant advances in the understanding of immunological and genetic factors. Numerous UC-associated genetic loci are in non-coding regions of the genome, and several are associated with lncRNAs [5].

Recently, the expression of two lncRNAs, *CDKN2B-AS1* and *GATA6-AS1*, has shown a correlation to disease severity and patient outcomes in UC patients [20,21]. The identification and study of lncRNAs have been accelerated by the rapid development of high-throughput technologies and bioinformatics. Meta-analyses of publicly available datasets have revealed both disease-specific genes and pathways [22]. Meta-analyses which include differing populations and conditions can increase the generalizability of results, as well as identify potential sources of bias [23]. In some instances, combining samples may increase statistical power. This study aimed to identify common differentially expressed lncRNAs across a set of publicly available UC datasets after manual annotation. The study shows the variation in lncRNA expression between different sample locations and disease states, highlighting the difficulties in the meta-analysis of lncRNAs in differing UC datasets.

## 2. Materials and Methods

### 2.1. Selection of GEO Datasets and Samples

Datasets were downloaded from GEO (https://www.ncbi.nlm.nih.gov/geo/) accessed between 1 November 2023 and 12 December 2023. For differential expression analysis, nine datasets were selected (GSE109142, GSE128682, GSE206285, GSE87466, GSE92415, GSE107499, GSE47908, GSE16879, GSE59071) [24–32], as they fulfilled the following criteria: datasets contained clearly annotated active UC samples, and control samples and were generated from human colonic tissue biopsies. Datasets were deposited in the NCBI GEO database between 2009 and 2022 and contained a total of 1171 samples from UC patients and 168 controls (Table 1). UC samples were evaluated using different scoring systems across different datasets. Dataset GSE109142 used the pediatric ulcerative colitis activity index (PUCAI) score and Mayo endoscopy sub-score. Dataset GSE59071 employed the UC disease activity index (UCDAI) endoscopy sub-score. Datasets GSE206285 and GSE87466 used the Mayo score. Datasets GSE92415 and GSE47908 used the Mayo score and endoscopy sub-score. Dataset GSE16879 utilized the Mayo endoscopic sub-score along with the histological score for UC. Two datasets (GSE92415 and GSE206285) included samples from clinical trials. Two of the datasets (GSE16879 and GSE47908) were run using the Affymetrix Human Genome U133 Plus 2.0 Array (Thermo Fisher Scientific, Waltham, Mass, USA), and three datasets (GSE92415, GSE206285, and GSE87466) the Affymetrix HT HG-U133 + PM Array (Thermo Fisher Scientific, Waltham, Mass, USA). Dataset GSE109142 was generated by the Illumina HiSeq 2500 (Illumina, San Diego, Cal, USA), GSE128682 by NextSeq550 (Illumina, San Diego, Cal, USA), GSE59071 by Affymetrix Human Gene 1.0 ST Array (Thermo Fisher Scientific, Waltham, Mass, USA), and GSE107499 by Affymetrix Human Gene Expression Array (Thermo Fisher Scientific, Waltham, Mass, USA). All datasets used in this study had PubMed identifiers except GSE107499, although this dataset was recently mentioned in Wu et al., in which lesional samples were assigned to active UC and non-lesional samples were assigned to controls [29]. Biopsy samples from patients with UC were reported as originating from various locations including the ascending colon, descending colon, the sigmoid colon or rectum, cecum, the edge of an ulcer or the most inflamed colonic segment, and 15 to 20 cm from the anal verge. Different methods were used for biopsy preservation including RNAlater, snap frozen in liquid nitrogen, formalin-fixed, and paraffin-embedded (FFPE), or the method was not reported in four datasets (Table 1).

**Table 1.** An overview of datasets used for meta-analysis.

| GEO Accession Number | PMID (Year) | UC Samples (N); (M/F) | Control Samples (N); (M/F) | Tissue | Platform | SSM |
|---|---|---|---|---|---|---|
| GSE109142 | 30604764 (2018) | 206 (112/94) | 20 (9/11) | rectal mucosal biopsy | Illumina HiSeq 2500 | NR |
| GSE128682 | 32322884 (2020) | 14 (9/5) | 16 (11/5) | sigmoid colon | NextSeq 550 | NR |
| GSE206285 | 36192482 (2022) | 550 (350/200) | 18 (9/9) | sigmoid colon | Affymetrix HT HG U133 + PM array | FFPE |
| GSE87466 | 29401083 (2018) | 87 (44/43) | 21 | 15–20 cm from anal verge | Affymetrix HT HG U133 + PM array | RNAlater |
| GSE92415 | 23735746 (2018) | 162 | 21 | colonic mucosal samples | Affymetrix HT HG U133 + PM array | NR |
| GSE107499 | NA (2018) | 59 (lesional) | 40 (non-lesional) | colon biopsy | Affymetrix Human Gene Expression Array | RNAlater |
| GSE47908 | 25358065 (2014) | 45 (20/25) | 15 (4/11) | descending colon | Affymetrix Human Genome U133 Plus 2.0 Arrays | RNA later/FFPE |
| GSE16879 | 19956723 (2009) | 24 (14/10) | 6 | colon | Affymetrix Human Genome U133 Plus 2.0 Arrays | NR |
| GSE59071 | 261692 (2015) | 97 | 11 | sigmoid or rectum | Affymetrix Human Gene 1.0 ST Array | snap-frozen |

NA = not available; NR = not reported; F = female; M = male; N = number of samples; FFPE = formalin-fixed paraffin-embedded tissue; SSM = sample storage method.

### 2.2. Dataset Curation

Samples from patients with active UC and control samples were manually selected based on information provided in the GEO database and corresponding publications. Samples that were excluded and not used for differential analysis included remission samples from dataset GSE128682. A full overview of the classification of the active UC vs. control samples for each of the nine datasets can be seen in Table S1.

### 2.3. Data Processing

The series matrix files for each dataset were downloaded from GEO. In cases where the datasets did not provide a normalized count matrix, the R DEseq2 package was used to perform normalization (GSE128682 and GSE48958) from the raw count matrix. The R edgeR (version 4.0.16) package was used to find differentially expressed lncRNA genes for active vs. control (Table S1) in each of the nine selected datasets. R packages, geneknitR (version 1.2.5) and gprofiler (version 0.2.3), were used to translate matrix IDs to symbol, Entrez, and Ensembl IDs. Cluster profiler (version 4.10.1) bitr function was used to identify ncRNAs by genetype filter [33]. Only lncRNAs with an EdgeR *p*-value less than 0.05 were considered significant. The results were combined to identify common differentially expressed lncRNAs across the datasets. Only the lncRNAs that were significantly differentially expressed in at least 33% of datasets (3 out of 9) were considered. A thirty-three percent cutoff was chosen by a Fisher test [34]. Given that approximately 5% of all transcripts were differentially expressed on average in all datasets, the chances of any transcript being expressed in 3 out of 9 datasets were unlikely (*p*.value 0.06); 4 or more gives a *p*-value less than 0.05.

### 2.4. Expression of lncRNAs in Different Disease States and Tissue Locations

The identified meta-signature lncRNAs using nine data sets were further examined in different disease states and locations of tissue across these datasets. A detailed description of all datasets can be found in Table S1. A *t*-test was employed to assess whether there is a statistically significant difference in lncRNA expression between disease states (Figure S1).

### 3. Results

#### 3.1. The Number of Annotated LncRNA Gene Symbols Found in Each Dataset

The number of lncRNA annotated gene symbols per dataset is depicted in Table 2. However, the number of lncRNAs found varies significantly from 4910 in dataset GSE128692 to 443 in GSE107499.

**Table 2.** Number of lncRNAs found per GEO dataset.

| Datasets * | LncRNAs # |
|------------|-----------|
| GSE107499 | 443 |
| GSE109142 | 2096 |
| GSE128682 | 4910 |
| GSE16879 | 2181 |
| GSE206285 | 2407 |
| GSE47908 | 2844 |
| GSE59071 | 778 |
| GSE87466 | 2843 |
| GSE92415 | 631 |

* Refers to the GEO series identifiers, **#** represents the total number of gene symbols that were annotated as "non-coding".

### 3.2. Common LncRNA Gene Symbols Found in One to Nine Matrices

The total number of lncRNA annotated gene symbols found represented in at least one of the nine datasets was 2416, for two datasets 1473, for three datasets 574, for four datasets 486, for five datasets 528, for six datasets 636, for seven datasets 248, and for eight datasets 148. The number of common lncRNA gene symbols found in all and nine datasets was 81.

### 3.3. Differentially Expressed lncRNAs

In this study, 19 lncRNAs have been identified as significantly differentially expressed, including 12 downregulated lncRNAs: CDKN2B antisense RNA (*CDKN2B-AS1*), DIP2C antisense RNA (*DIP2C-AS1*), DPP10 antisense RNA (*DPP10-AS1*), FOXD2 adjacent opposite strand RNA (*FOXD2-AS1*), GATA6 antisense RNA (*GATA6-AS1*), microRNA 215 (*MIR215*, *MIR3936HG*), long intergenic non-protein coding RNA 1224 (*LINC01224*), long intergenic non-protein coding RNA 2023 (*LINC02023*), SATB2 antisense RNA (*SATB2-AS1*), TP53 target 1 (*TP53TG1*), VLDLR antisense RNA (*VLDLR-AS1*). Seven lncRNAs were upregulated in active UC including: colorectal neoplasia differentially expressed (*CRNDE*), family with sequence similarity 30 member A (*FAM30A*), uncharacterized LOC643977 (*FLJ32255*), long intergenic non-protein coding RNA 1215 (*LINC01215*), long intergenic non-protein coding RNA 3040 (*LINC03040*), myocardial infarction associated transcript (*MIAT*), MIR155 host gene (*MIR155HG*). Each of these nineteen lncRNAs were differentially expressed in at least three out of the nine datasets. Which differentially expressed lncRNA was found in which dataset is shown in Table 3.

The expression levels of the lncRNAs were compared across different disease states depicted in Table 3, revealing several significant differentially expressed lncRNAs. An example of a boxplot depicting the pairwise comparison of lncRNA expression in different disease states can be seen in Figure 1.

Boxplots showing the expression patterns of all lncRNAs in different disease states can be found in Figure S1.

The expression levels of lncRNAs were also compared across tissue locations. Variations in the expression levels of lncRNAs among tissue locations within the same disease state are shown in an example plot (Figure 2). Boxplots for each lncRNA across annotated tissue locations are shown in Figure S2. For completeness, datasets that were excluded from the analysis, GSE38713, GSE48634, GSE9452, GSE38713, GSE48958, and GSE55306, are also included in Figure S2.

**Table 3.** The candidate lncRNAs in each GEO dataset.

| LncRNA | GSE107499 | GSE10942 | GSE128682 | GSSE16879 | GSE206285 | GSE47908 | GSE59071 | GSE87466 | GSE92415 | sig_pct | nmat |
|---|---|---|---|---|---|---|---|---|---|---|---|
| *MIR215* | N | S | S | N | N | N | S | N | N | 100 | 3 |
| *DPP10-AS1* | N | S | S | Y | S | S | N | S | N | 83.3 | 6 |
| *FAM30A* | S | S | Y | S | S | S | Y | S | S | 77.8 | 9 |
| *LINC02023* | N | N | N | Y | S | S | N | S | N | 75 | 4 |
| *MIR155HG* | N | S | S | N | N | Y | N | N | S | 75 | 4 |
| *CDKN2B-AS1* | Y | S | S | Y | S | Y | N | S | S | 62.5 | 8 |
| *VLDRL-AS1* | N | S | S | N | N | Y | Y | S | N | 60 | 5 |
| *MIAT* | N | S | Y | Y | S | Y | N | S | S | 57.1 | 7 |
| *CRNDE* | S | S | Y | Y | Y | N | N | N | S | 50 | 6 |
| *FLI32255* | N | N | Y | Y | S | Y | N | S | S | 50 | 6 |
| *GATA-AS1* | N | S | Y | Y | S | Y | N | S | N | 50 | 6 |
| *LINC01215* | N | S | S | Y | Y | Y | N | S | N | 50 | 6 |
| *LINC01224* | N | S | Y | Y | S | Y | N | S | N | 50 | 6 |
| *MIR3936HG* | N | N | Y | Y | S | Y | N | S | S | 50 | 6 |
| *SATB2-AS1* | Y | S | Y | Y | S | Y | S | S | N | 50 | 8 |
| *DIP2C-AS1* | Y | S | Y | Y | S | N | Y | N | S | 42.9 | 7 |
| *FOXD2-AS1* | N | S | Y | Y | Y | Y | N | S | S | 42.9 | 7 |
| *LINC03040* | S | S | S | Y | Y | Y | Y | Y | Y | 33.3 | 9 |
| *TP53TG1* | Y | S | Y | Y | Y | Y | Y | S | S | 33.3 | 9 |

N = lncRNA not present in the dataset; Y = lncRNA present in the dataset; S = LncRNA significantly differentially expressed in the dataset; nmat = number of datasets; sig pct = significant percentage.

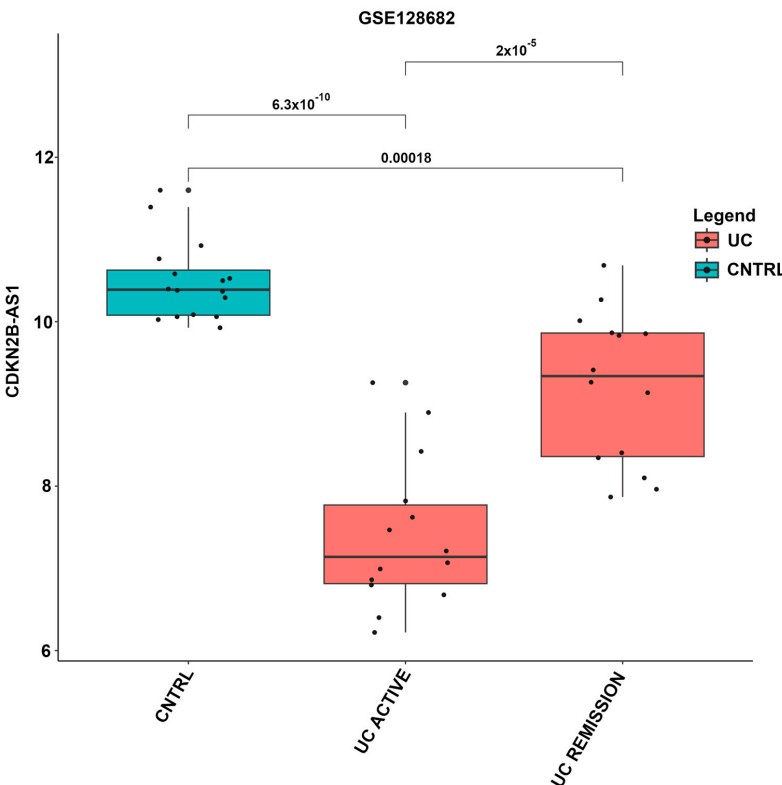

**Figure 1.** Boxplot of expression levels of lncRNA *CDKN2B-AS1* in different UC disease states. Expression values and disease state were taken from the GSE128682 dataset and annotation. The x-axis

represents the annotated disease states, including control, active UC, and UC in remission. Boxplots containing control samples are indicated in blue, and UC active and remission samples in red. The y-axis indicates *CDKN2B-AS1* expression levels, where each black dot represents an individual sample. The *p*-values for each disease state comparison are indicated above the boxplots.

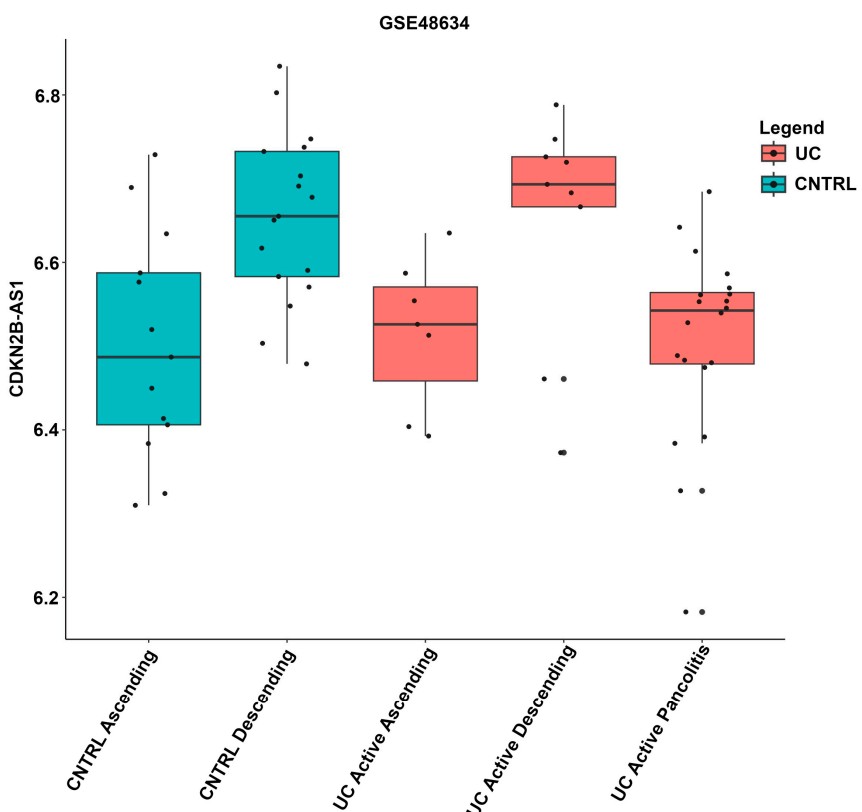

**Figure 2.** Boxplot of lncRNA *CDKN2B-AS1* expression in distinct tissue locations. Expression values, disease state, and tissue location were taken from the GSE48634 dataset and annotation. The x-axis indicates the annotated tissue location. Boxplots containing active UC samples are shown in red, non-IBD controls are indicated as blue. The y-axis indicates *CDKN2B-AS1* expression levels, where each black dot represents an individual sample.

## 4. Discussion

This study highlights the challenges related to performing a lncRNA meta-analysis on a complex disease such as UC. In the publicly available datasets, both the description of the UC disease state and location of the colonic biopsy location differ. UC disease states annotated in the different datasets include active, inactive, macroscopic inflammation, and remission, which may exhibit varying levels of inflammation and were shown to have an influence on lncRNA transcription levels. In this study, the expression of lncRNA *CDKN2B-AS1* was significantly downregulated in UC compared to controls but significantly upregulated in UC remission compared to active UC (Figure 1). Grouping UC remission along with active UC samples would reduce the probability of identifying *CDKN2B-AS1* as differentially expressed especially after multiple correction. Several lncRNAs exhibited significantly different expression levels across various disease states in this study (Figure S2).

Sample metadata varied significantly among GEO datasets. Information about tissue biopsy location, medication, gender, and age were not listed in some datasets. Different tissue locations have been shown to influence lncRNA expression profiles [35,36]; unfortunately, subgrouping by available tissue location would lead to groups that were too small for a robust statistical analysis. Comparison of lncRNA expression between tissue types

could lead to erroneous interpretations depicted in Figure 2. A recent review of lncRNA mucosal transcripts implicated in UC, Crohn's disease, and celiac disease revealed that the lncRNAs showed significantly more location-specific expression along the GI tract than the protein-coding genes [36]. Comparing tissue types directly could lead to a more comprehensive set of tissue-specific differentially expressed lncRNAs in UC. However, this study identified lncRNAs that are differentially expressed to a varying extent in several colonic tissues. These lncRNAs may be associated with common but not tissue-specific processes such as inflammation.

This study acknowledges tissue-specific lncRNA expression, as shown in Figure S2. The boxplots show substantial variation in tissue specific lncRNA expression levels in both UC and control groups. For example, in dataset GSE107499, the expression levels of *DIP2C-AS1* in lesional (active UC) cecum samples were like the controls, whereas other tissue locations showed a downregulation of *DIP2C-AS1* (Figure S3). It has been shown that lncRNA expression can vary depending on biopsy tissue location within the large intestine [37]. However, some previous meta-analysis studies have not taken biopsy tissue location into account [38,39].

The comparison of lncRNA expression between datasets is challenging as the same lncRNA may be represented by different gene symbols in different datasets [40]. Therefore, the R packages geneknitR and gprofiler were utilized to deal with the lack of consistency in gene symbol identifiers [29] These tools enabled the translation of count matrix IDs into symbols, Entrez, and Ensembl IDs. The Entrez identifiers were utilized by the cluster profiler bitr function for verifying gene symbols and potential aliases, as well as identifying ncRNAs by gene type. This approach is conservative, and some lncRNAs were lost in the gene symbol translation process. The inclusion of microarray data presents further challenges. Prior to the use of RNAseq, microarrays were a commonly used transcriptomic methodology, and a lot of valuable microarray results remain available in genomic databases. Unfortunately, the information provided by microarray experiments is limited to the design of the chip. Microarrays are primarily designed to detect and quantify protein-coding genes; consequently, many lncRNAs are not included in early microarray platforms [41]. Unlike RNAseq, microarray results cannot be realigned to current genomes.

While 4910 lncRNAs were found from sequencing dataset GSE128682, only 443 could be identified from human gene expression array dataset GSE107499 (Table 3). Therefore, the number of lncRNA identifiers present in all datasets decreased as more datasets were included. An additional challenge is the current lack of consensus regarding the total number of defined lncRNAs [10]. Therefore, the identification of specific lncRNAs depends on which database was used for annotation.

Manual curation is a key step in identifying differentially expressed genes in publicly available datasets, as the metadata associated with gene expression studies within GEO typically do not adhere to controlled vocabularies to describe biological entities such as tissue type, cell type, cell line, gene identifiers, treatment, and disease. For example, comparing all UC labeled samples without removing inactive UC samples from each dataset would result in a different result. The annotation of genes varied in all nine GEO datasets. Only a few commonly differentially expressed lncRNAs across independent UC datasets were found, even after manual curation, clearly showing the challenges in comparing data sets.

Nineteen lncRNAs were identified that were differentially expressed between active UC and controls in at least three datasets of the nine GEO datasets. Of these nineteen lncRNAs, *miR-215*, *FOXD2-AS1*, *SATB2-AS1*, *TP53TG1*, *LINC01224*, *CRNDE*, and *DPP10-AS1* have been implicated in colorectal cancer (CRC) [42–48]. The higher expression of these lncRNAs may be associated with promoting colorectal cancer (CRC) through regulating gene expression, epithelial to mesenchymal transition (EMT), cell cycle progression, and by promoting tumor proliferation, invasion, and migration.

The long non-coding RNA colorectal neoplasia differentially expressed (*CRNDE*) was found to be upregulated in UC (Figure S2). Its overexpression and potential role in tumori-

genesis in CRC have been reported in several studies [49,50]. Therefore, monitoring *CRNDE* expression in UC patients may serve as a predictive biomarker for identifying individuals with UC at risk of developing cancer. In addition to the lncRNAs discussed above, this study identified several differentially expressed lncRNAs that have been previously characterized as dysregulated in UC. These include the following lncRNAs: *CDKN2B-AS1*, *DPP10-AS1*, *FOXD2-AS1*, *MIR155HG*, *MIAT*, and *GATA6-AS1* [5,20,21,51,52]. The expression pattern of these lncRNAs is consistent with our findings (Figure S2). LncRNAs *CDKN2B-AS1*, *CRNDE*, *DPP10-AS1*, and *GATA6-AS1* have been studied in the context of UC, with documented roles in various functions, including maintaining intestinal barrier integrity and modulating inflammation during the progression of UC [5,20,36,48]. A recent study has demonstrated an association between reduced *GATA6-AS1* expression and increased UC severity, as well as an unfavorable clinical outcome. They also highlighted the potential contribution of *GATA6-AS1* in regulating mitochondrial respiration, suggesting its involvement in maintaining epithelial integrity and gastrointestinal pathology [21]. *CDKN2B-AS1* has been shown to correlate with disease severity and UC progression by regulating proliferation, apoptosis, barrier function, and inflammation response in colon cells [20]. Interestingly, when found, lncRNA *CDKN2B-AS1* was differentially expressed in 62% of datasets, and *GATA6-AS1* (50%).

In addition to the CRC associated lncRNAs, many of the differentially regulated lncRNAs have been previously characterized in UC. These include lncRNAs *CDKN2B-AS1*, *DPP10-AS1*, *FOXD2-AS1*, *MIR155HG*, *MIAT*, and *GATA6-AS1*. The observed expression patterns of these lncRNAs are found to be consistent with previous findings [5,20,21,48,52].

## 5. Conclusions

The lncRNAs were present and differentially expressed in several human UC GEO datasets and could represent general markers for active UC independent of biopsy location, age, gender, and treatment. Several of the lncRNAs are associated with CRC and could potentially be used as clinical indicators for monitoring CRC risk in ulcerative coli-tis patients. Promising molecular biomarkers, lncRNAs, have the potential to enhance the accuracy, sensitivity, and specificity of molecular methods employed in clinical diagnosis. In standard medical practice, the development of lncRNA-based diagnostics and therapies will be helpful to improve patient clinical care and quality of life [53]. However, some of the challenges of analyzing publicly available independent UC datasets remain. Significant manual annotation will remain a key step in the comparative analysis of UC datasets.

**Supplementary Materials:** The following supporting information can be downloaded at: https://www.mdpi.com/article/10.3390/cimb46040198/s1.

**Author Contributions:** C.G.F.: data curation, conceptualization, methodology, investigation, visualization, validation, software, writing, review and editing. M.K.R.: formal analysis, validation, writing, reviewing the final draft. R.H.P.: conceptualization, investigation, validation, project administration, resources, methodology, supervision, writing, review and editing. All authors have read and agreed to the published version of the manuscript.

**Funding:** This research received no external funding.

**Institutional Review Board Statement:** Not applicable.

**Informed Consent Statement:** Not applicable.

**Data Availability Statement:** All data generated or analyzed during this study are included in this published article and Supplementary Materials.

**Conflicts of Interest:** The authors declare no conflicts of interest.

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
