# Peer review of "Challenges in Defining a Reference Set of Differentially Expressed lncRNAs in Ulcerative Colitis by Meta-Analysis"

_cimb, doi:10.3390/cimb46040198_

Round 1
Reviewer 1 Report
Comments and Suggestions for Authors
In the present study, which utilized publicly available databases, the authors aim to identify lncRNAs associated with ulcerative colitis (UC). I appreciate the efforts made in this research and would like to offer some suggestions and pose a few questions to potentially enhance the manuscript's quality and impact.
1. The association between lncRNAs and UC appears tenuous based on the current exposition. I recommend that the authors expand both the Introduction and Discussion sections to address the following points: 1) the established roles of lncRNAs in clinical settings, including their utility in diagnosis, treatment, and prognostication; 2) the existing challenges in diagnosing and treating UC in clinical practice; 3) the rationale for investigating differentially expressed lncRNAs in the context of UC and how this could potentially refine diagnostic and therapeutic strategies; and 4) the implications of the lncRNAs identified in this study for the diagnosis and treatment of UC, elucidating how these findings might translate into clinical advancements.
2. The manuscript should clarify the current gold standard for UC diagnosis and the most reliable tissue biopsies used in clinical practice. The inclusion of various tissue types in Table 1 raises questions about the rationale and necessity for such diversity, especially considering the tissue-specific expression patterns of lncRNAs. This aspect could lead to erroneous interpretations, and I urge the authors to provide a clear justification for their tissue selection strategy.
3. The Discussion section offers limited information on the 19 lncRNAs identified in the current study. I suggest that the authors provide more information of the existing knowledge on these lncRNAs, including any known or potential associations with UC or other pathologies. Expanding on this discussion could elucidate potential mechanisms through which these lncRNAs might influence UC pathogenesis or progression, thereby reinforcing the study's relevance and contributing to the field.
Comments on the Quality of English LanguageThe use of abbreviation need to be revised. The abbreviation should be defined when the full name first appeared in the main content.
Author Response
The responses to the reviewer can be found in the file attached.

Reviewer 2 Report
Comments and Suggestions for Authors
Major comments:
1. Apparently the different datasets used for this study had been referring to different versions of the human genome. Therefore, it is advisable to perform a re-analysis based on raw data files with the most recent version of the human genome and not on processed data.
2. Table 2: The rather different number of lncRNAs found in the different datasets suggest that they vary quite a lot in quality. This should be corrected.
Minor comments:
1. Gene name abbreviations should be in italic. This applies also to Tables.
Comments on the Quality of English LanguageOnly minor corrections needed.
Author Response
- Apparently the different datasets used for this study had been referring to different versions of the human genome. Therefore, it is advisable to perform a re-analysis based on raw data files with the most recent version of the human genome and not on processed data.
Two of the datasets were RNASeq, GSE128682 was aligned against gencode V28 (150bp paired end), GSE109142 against gencode 25 (75 bp paired end). Realigning 4 random samples from GSE109142 against gencode version 28 did not provide any new lncRNAs. Note that read coverage of GSE128682 was significantly higher than GSE109142. The differences between v25 and v28 are not major as both are built upon hg38. The major limiting factor was the microarray data. Before the widespread use of RNASeq, microarray was the most common technique used to explore gene expression. Microarrays are collections of DNA probes fixed to a substrate. Microarray “raw data” is the intensity of hybridization of labeled probes to these fixed probes. If no probes exist for a particular lncRNA then no amount of reanalysis will help. Nor can one just realign fastq files against the current genome build, as would be done in RNASeq experiments. Packages such as geneknitR attempt to ensure that the gene annotation (correct at the time of microarray chip design) can be converted to current annotation. Fastq files for all samples would be ideal. However, there is a large amount of valuable information available in genome archives from microarray experiments. They cannot be realigned but still represent a valuable resource.
- Table 2: The rather different number of lncRNAs found in the different datasets suggest that they vary quite a lot in quality. This should be corrected.
The low number of identified lncRNAs reflects the chip probe design of the microarrays used. As mentioned above there is no way to correct missing probes.
Minor comments:
- Gene name abbreviations should be in italic. This applies also to Tables.
Genes are now in italic.
Round 2
Reviewer 2 Report
Comments and Suggestions for Authors
The manuscript improved, but the technical limitations, as replied by the authors to my main point, should be discussed in the manuscript.
Comments on the Quality of English LanguageThe English is OK
Author Response
The manuscript improved, but the technical limitations, as replied by the authors to my main point, should be discussed in the manuscript.
Limitations of microarray technology is now mentioned in the discussion , line 244-247, and line 249-250 (indicated in red).
